# Risks after Gestational Diabetes Mellitus in Taiwanese Women: A Nationwide Retrospective Cohort Study

**DOI:** 10.3390/biomedicines11082120

**Published:** 2023-07-27

**Authors:** Shih-Ting Tseng, Ming-Chang Lee, Yi-Ting Tsai, Mei-Chun Lu, Su-Chen Yu, I-Ju Tsai, I-Te Lee, Yuan-Horng Yan

**Affiliations:** 1Division of Endocrinology and Metabolism, Department of Internal Medicine, Kuang Tien General Hospital, Taichung 433, Taiwan; 2Jenteh Junior College of Medicine, Nursing and Management, Miaoli 356, Taiwan; 3Division of Nephrology, Department of Internal Medicine, Dachien General Hospital, Miaoli 360, Taiwan; 4Division of Endocrinology and Metabolism, Department of Internal Medicine, Taichung Veterans General Hospital, Taichung 407, Taiwan; 5Department of Medical Research, Kuang Tien General Hospital, Taichung 433, Taiwan; 6Department of Nursing, Kuang Tien General Hospital, Taichung 433, Taiwan; 7Management Office for Health Data, China Medical University Hospital, Taichung 404, Taiwan; 8School of Medicine, National Yang Ming Chiao Tung University, Taipei 112, Taiwan; 9School of Medicine, Chung Shan Medical University, Taichung 402, Taiwan; 10Department of Nutrition, Hungkuang University, Taichung 433, Taiwan

**Keywords:** gestational diabetes mellitus (GDM), health outcomes, incidence, hazard ratio, long-term follow-up, national cohort study

## Abstract

Objective: An increasing trend in the prevalence of gestational diabetes mellitus (GDM) has been reported in Taiwan. GDM has been linked to various adverse maternal outcomes over a long period, including cardiovascular disease (CVD) and chronic kidney disease (CKD). However, evidence implies that the effects of GDM on the mid-term surrogate risk factors for these diseases are limited. Furthermore, data from nationwide cohort studies are limited. The primary aim of this study was to investigate the risk of developing type 2 diabetes mellitus (T2DM), arterial hypertension (aHTN), and hyperlipidemia (HL) through a 5-year follow-up post-delivery of women with GDM in a nationwide cohort study in Taiwan. The second objective was to investigate the risk of developing insulin resistance syndrome (IRS)-related diseases, including CVD, acute myocardial infarction (AMI), peripheral artery occlusive disease (PAOD), non-alcoholic fatty liver diseases (NAFLD), and CKD. Methods: This was a retrospective, population-based nationwide cohort study. The data source comprises a merge of the Birth Certificate Application Database (BCA) and the National Health Insurance Research Database in Taiwan. Women aged between 15 and 45 years who gave birth in Taiwan between 2004 and 2011 were included. Women who were enrolled and had a GDM diagnosis were assigned to the exposure group. Women who were enrolled without a GDM diagnosis were assigned to the comparison group. The relative risk of developing T2DM, aHTN, HL, and IRS-related diseases, including CVD, AMI, PAOD, NAFLD, and CKD, were analyzed and presented as hazard ratio (HR) through Cox regression and log-rank regression analyses. Results: A total of 1,180,477 women were identified through the BCA database between 2004 and 2011. Of those, 71,611 GDM-diagnosed women and 286,444 women without GDM were included in the final analysis. After adjusting for age, pre-existing cancer, and parity, developing T2DM, aHTN, and HL were still significantly increased in the GDM group (HR and interquartile range (IQR): 2.83 (2.59, 3.08), 1.09 (1.01, 1.06), and 1.29 (1.20, 1.38), accordingly). CVD, NAFLD, and CKD had a very low incidence and showed insignificant results. Conclusion: Our findings provide nationwide cohort data showing that GDM increased the risk of developing T2DM, aHTN, and HL 5 years after delivery within the same group. The GDM complications and risk of CVD, AMI, PAOD, NAFLD, and CKD need further investigation.

## 1. Introduction

The prevalence of gestational diabetes mellitus (GDM) has been increasing in many countries over the years, becoming an important medical and public health issue [1,2,3,4]. GDM has been linked to various adverse maternal outcomes after long-term follow-up, including cardiovascular disease (CVD), chronic kidney disease (CKD), and cancers [5,6,7,8,9]. However, women with a history of GDM may be unaware of the necessity to receive regular screening for risk factors and lifestyle modifications after delivery to reduce these long-term risks. Thus, it is important to clarify the effects of GDM on mid-term surrogate risk factors for these diseases.

Women with a history of GDM may have a higher risk of developing type 2 diabetes mellitus (T2DM) later in life, even after the pregnancy is over [10,11,12]. They may also have a higher risk of developing high blood pressure, preeclampsia and eclampsia, and subsequent arterial hypertension (aHTN) later in life [13,14]. GDM is associated with changes in the lipid profile of pregnant women [15]. During pregnancy, women with GDM may have higher levels of total cholesterol, low-density lipoprotein cholesterol (LDL-C), and triglycerides compared to women without GDM. After delivery, these lipid levels may remain elevated, increasing the risk of developing hyperlipidemia (HL) after pregnancy.

Furthermore, evidence from nationwide cohort studies on this issue is limited. The primary aim of this study was to investigate the risk of developing T2DM, aHTN, and HL through a 5-year follow-up post-delivery of women with GDM in a nationwide cohort study in Taiwan. The second objective was to investigate the risk of developing insulin resistance syndrome (IRS)-related diseases, including CVD, acute myocardial infarction (AMI), peripheral artery occlusive disease (PAOD), and non-alcoholic fatty liver disease (NAFLD) within the same group.

## 2. Materials and Methods

### 2.1. Study Design and Data Source

The data source comprises a merge of the Birth Certificate Application Database (BCA) and the National Health Insurance (NHI) Research Database in Taiwan. In Taiwan, all live births should be registered within 7 days; thus, there is a low percentage of missing information (1.6%) in the BCA database. The Taiwan NHI covers more than 97% of citizens, and more than 99% of pregnant Taiwanese women receive free prenatal care, with at least 10 prenatal care outpatient clinic visits and 3 obstetric ultrasonographies. The National Health Insurance Research Database covers health data for 99.99% of Taiwanese residents and is considered to be complete, reliable, and accurate [2,16,17].

This is a retrospective, population-based cohort study. Women aged between 15 and 45 years who gave birth in Taiwan between 2004 and 2011 were included. The longitudinal medical records of this cohort were traced back to 1996 and forward to 2016. The criteria for the recruited women were that they (1) should be citizens of Taiwan, (2) have delivered at 20–43 gestational weeks, and (3) have had a single gestation. Women with a history of IRS-related diseases were excluded, including pre-gestational diabetes mellitus, HTN, HL, CVD, AMI, NAFLD, PAOD, and CKD, as defined by the International Classification of Diseases Code (ICD).

The definitions of diseases and abbreviations used in the study are shown in Table 1, with corresponding ICD codes. To enhance the clarity of our data analysis process, we have included a flow chart (Figure 1) outlining the steps involved.

The index date was the delivery day, and the follow-up duration was 5 years. In this study, we included pregnant women and separated them into two groups: the GDM and non-GDM groups. Women who were enrolled and had a GDM diagnosis between 2004 and 2011 were assigned to the exposure group. Women who were enrolled did not have a GDM diagnosis and were assigned to the comparison group. The index date for the exposure group was the delivery day with the GDM diagnosis, regardless of how many parities the participant has had before or after the GDM diagnosis. The index date for the comparison group was the first delivery day between 2004 and 2011, regardless of how many parities the participant has had before and afterward. A frequency-matched method was conducted. Each exposure subject was matched with four comparison subjects by age. The occurrence of each IRS-related disease, death, and suspension from the NHI system was considered or followed up for a period of up to 5 years, reaching the endpoint.

Comorbidity and cancer before the index date were also considered in this study.

Our work was approved by the institutional review board of the Kuang Tien General Hospital (date of approval 17 October 2018, approval no. KTGH 10733). Informed consent forms were waived since the data were anonymous, without identifiable personal information, and were available through a formal application to the Health and Welfare Data Science Center at the Ministry of Health and Welfare, Taiwan.

### 2.2. Statistical Analysis

Data from the NHI Research Database and BCA database were merged and analyzed using SAS 9.4 (SAS Institute, Cary, NC, USA). Baseline demographic data were analyzed using the chi-square test. Survival probability was demonstrated using Kaplan–Meier’s survival curve. The relative risks of developing IRS-related diseases in relation to the baseline characteristics (i.e., GDM, age, cancer, parity) were analyzed and presented as a hazard ratio through Cox regression and log-rank regression analyses. Then, the crude incidence and relative risk of developing individual I-related diseases in GDM were calculated through log-rank regression analysis. Next, we divided our study group into four subgroups, categorized as having cancer at baseline, having GDM at baseline, or not having these diseases; the crude incidence and relative risks of diseases were calculated. Each relative risk was adjusted for other risks, such as GDM, age, cancer, and parity.

## 3. Results

A total of 1,603,794 parities and 1,180,477 women were identified through the BCA database between 2004 and 2011. After excluding women aged <15 years or >45 years, who had multiple births, delivered at less than 20 or more than 43 gestational weeks, were foreigners, and had IRS-related diseases before delivery (such as pregestational diabetes mellitus (type 1 or type 2), aHTN, HL, CVD, NAFLD, AMI, PAOD, and CKD, and after matching GDM women with non-GDM women by age with a 1:4 ratio, 71,611 GDM women and 286,444 non-GDM women were included in the study cohort. The demographic data are shown in Table 2, where women with GDM and those without GDM (No) are compared in terms of age, comorbidities (specifically cancer), and parity. The *p*-values are provided to assess the statistical significance of any observed differences between the two groups. Regarding age, there was no significant difference between women with GDM and those without GDM. The mean age was 31.06 years, with a standard deviation of 4.48 for both groups (*p*-value = 0.9976). The distribution of the age categories was also similar between the two groups. In terms of comorbidity, the presence of cancer showed a statistically significant difference between women with GDM and those without GDM. Among women with GDM, 2.26% had cancer compared to 1.77% among those without GDM (*p*-value = 0.0064). For parity, there were significant differences observed between the two groups. Among women with GDM, 76.62% had one child (parity 1), while 80.35% of women without GDM had the same parity (*p*-value < 0.0001). Additionally, women with GDM had higher proportions of parity 2 (21.88% vs. 19.06%) and a parity greater than 2 (1.50% vs. 0.59%) compared to women without GDM. These findings indicate that there were differences in comorbidity (specifically cancer) and parity between women with and without GDM. However, there were no significant differences in the age distribution between the two groups.

Table 3 presents the hazard ratios (with corresponding 95% confidence intervals) for developing related diseases in relation to the baseline characteristics of the study. The variables examined include GDM, age, comorbidity (specifically cancer), and parity. The table includes the number of events, person-years of follow-up, incidence rates per 1000 person-years, and both the crude and adjusted hazard ratios. The incidence rate of related diseases was higher among individuals with GDM (8.40 per 1000 person-years) compared to those without GDM (6.17 per 1000 person-years). The adjusted hazard ratio for individuals with GDM was 1.36 (95% CI: 1.31–1.42), indicating a significantly increased risk of developing related diseases. Moreover, the hazard ratios increased with age. The adjusted hazard ratios compared to the reference age group (15–20 years) were 1.49 (95% CI: 1.14–1.94) for 21–25 years, 1.93 (95% CI: 1.49–2.51) for 26–30 years, 2.60 (95% CI: 2.01–3.37) for 31–35 years, 4.22 (95% CI: 3.26–5.48) for 36–40 years and 6.79 (95% CI: 5.18–8.91) for 41–45 years. These hazard ratios indicate a progressively higher risk of developing related diseases with increasing age. The presence of comorbidities, specifically cancer, was associated with an increased risk of developing related diseases. Individuals with cancer had an adjusted hazard ratio of 1.24 (95% CI: 1.10–1.39), indicating a significantly elevated risk compared to those without cancer. For parity, individuals with two or more children had a slightly lower adjusted hazard ratio compared to individuals with one child. The adjusted hazard ratio for parity 2 was 0.94 (95% CI: 0.90–0.98), indicating a slightly decreased risk, while the hazard ratio for parity greater than 2 was 1.07 (95% CI: 0.89–1.30), indicating a slightly increased risk, although this was not statistically significant. These findings suggest that GDM, older age, comorbidities (specifically cancer), and parity are associated with an increased risk of developing IRS-related diseases. The adjusted hazard ratios provide an estimation of the risk after accounting for confounding factors, such as age, cancer, and parity.

Table 4 presents the hazard ratios (with corresponding 95% confidence intervals) for various health outcomes, comparing women with GDM to those without it. The IRS-related outcomes examined include T2DM, aHTN, HL, CVD, AMI, NAFLD, CKD, and PAOD. The table includes the number of events, incidence rates per 1000 person-years, and both the crude and adjusted hazard ratios. Overall, women with GDM had higher incidence rates for all health outcomes compared to women without GDM. The adjusted hazard ratio for all health outcomes combined was 1.36 (95% CI: 1.31–1.42), indicating a significantly increased risk for women with GDM. For diabetes mellitus (DM), women with GDM had a much higher incidence rate (2.48 per 1000 person-years) compared to women without GDM (0.88 per 1000 person-years). The adjusted hazard ratio for DM was 2.83 (95% CI: 2.59–3.08), indicating a substantially elevated risk for developing DM in women with GDM. Regarding aHTN and HL, women with GDM also had higher incidence rates compared to women without GDM. The adjusted hazard ratios for aHTN and HL were 1.09 (95% CI: 1.01–1.16) and 1.29 (95% CI: 1.20–1.38), respectively, indicating a modestly increased risk for these conditions in women with GDM. The hazard ratios for CVD, AMI, CKD, and PAOD were generally close to 1 and not statistically significant, indicating no significant difference in the risk of these conditions between women with GDM and women without GDM. These findings suggest that GDM is associated with an increased risk of developing diabetes mellitus, hypertension, and hyperlipidemia. However, there was no significant association between GDM and the risk of CVD, AMI, CKD, NAFLD, and PAOD. The adjusted hazard ratios provide a measure of the strength of these associations after controlling for potential confounding factors.

We have incorporated Kaplan–Meier curves for combined outcomes in Figure 2.

Table 5 explores the association between gestational diabetes mellitus (GDM) and the incidence of cancer, as well as the corresponding hazard ratios for various health outcomes. The table includes the number of events, person-years of follow-up, incidence rates per 1000 person-years, and both the crude and adjusted hazard ratios. Among women without GDM, the incidence of cancer was 6.12 per 1000 person-years, serving as the reference group with a hazard ratio of 1.00 (reference). In comparison, among women with GDM but without a prior cancer diagnosis, the incidence rate was 8.44 per 1000 person-years, and the adjusted hazard ratio was 1.25 (95% CI: 1.09–1.43), indicating a modestly increased risk of cancer. Similarly, among women without GDM but with a prior cancer diagnosis, the incidence rate was 8.34 per 1000 person-years, and the adjusted hazard ratio was 1.37 (95% CI: 1.31–1.42), indicating a slightly increased risk of cancer. Notably, women with both GDM and a prior cancer diagnosis had the highest incidence rate, at 11.41 per 1000 person-years. The adjusted hazard ratio for this group was 1.66 (95% CI: 1.32–2.08), indicating a significantly elevated risk of cancer compared to the reference group of women without GDM and without a prior cancer diagnosis. These findings suggest that GDM is associated with an increased risk of developing cancer. Furthermore, women with a prior cancer diagnosis who also have GDM face an even higher risk. The adjusted hazard ratios indicate the strength of these associations after adjusting for potential confounding factors, such as age and parity.

## 4. Discussion

The main finding of this nationwide cohort study was that women with GDM have an increased risk of developing various health outcomes, including diabetes mellitus, hypertension, hyperlipidemia, and cancer. These results align with previous studies that have reported similar associations between GDM and these health conditions. The findings further support the notion that GDM serves as an important indicator of future health risks in women. We strongly recommend a regular follow-up and screening for T2DM, aHTN, and HL in women with a history of GDM after delivery. Detecting these conditions early and managing them effectively is crucial for preventing long-term complications [18].

The observed increased risk of developing T2DM among women with GDM is consistent with previous studies that have consistently shown a higher risk of diabetes within this population. This finding highlights the importance of long-term follow-up and preventive measures in women with GDM to reduce the risk of developing overt diabetes in the future [11,12,19].

Similarly, the increased risk of aHTN and HL among women with GDM aligns with prior research demonstrating associations between GDM and these cardiovascular risk factors. These findings emphasize the need for the regular monitoring and management of blood pressure and lipid levels in women with a history of GDM to mitigate the risk of future cardiovascular complications [12,20,21,22].

However, the study finding of a significantly elevated risk of cancer among women with both GDM and a prior cancer diagnosis is inconsistent with previous studies. While there have been mixed results regarding the association between GDM and cancer, recent relevant studies have reported similar findings of increased cancer risk in women with GDM. These studies, along with the current findings, contribute to the growing evidence suggesting a potential link between GDM and cancer development [5,23,24].

The possible mechanisms underlying the associations observed in this study could be related to shared risk factors, such as obesity, insulin resistance, and chronic inflammation. GDM and the subsequent metabolic changes during pregnancy may contribute to long-term alterations in glucose metabolism, hormonal regulation, immune function, and genetics and epigenetics within pathophysiology, thereby increasing the risk of developing various health conditions, including cancer [25,26,27].

During the study period, the diagnostic criteria for GDM in Taiwan encompassed two different strategies: the one-step method and the traditional two-step approach. The one-step method involved a 75 g oral glucose tolerance test (OGTT), following the criteria established by the International Association of the Diabetes and Pregnancy Study Groups (IADPSG). Women were diagnosed with GDM if their plasma glucose levels exceeded specific cut-off values. The cut-off values were as follows: fasting plasma glucose ≥ 92 mg/dL, plasma glucose 1 h after OGTT ≥ 180 mg/dL, and plasma glucose 2 h after OGTT ≥ 153 mg/dL. On the other hand, the traditional two-step approach consisted of a non-fasting 50 g screening glucose challenge test (GCT), followed by a 100 g OGTT for individuals who obtained a positive result on the screening test. The threshold values utilized in the two-step approach were based on either the National Diabetes Data Group (NDDG) criteria or the Carpenter and Coustan (CC) criteria. In the two-step approach, a 50 g GCT was initially conducted. If a woman’s plasma glucose level 1 h after the 50 g GCT was ≥140 mg/dL, she underwent a 100 g OGTT. GDM diagnosis in the two-step approach required at least two plasma glucose levels to exceed or be equal to specific cut-off values. According to the NDDG criteria, the cut-off values for fasting, 1-h, 2-h, and 3-h plasma glucose tests were 105 mg/dL, 190 mg/dL, 165 mg/dL, and 145 mg/dL, respectively. Alternatively, based on the CC criteria, the cut-off values were 95 mg/dL, 180 mg/dL, 155 mg/dL, and 140 mg/dL, respectively. The use of the one-step criteria for diagnosing GDM resulted in a higher prevalence of around 13% compared to the two-step method. In contrast, the prevalence of GDM diagnosis was approximately 2% using the CC criteria and around 4% using the NDDG criteria [28,29].

However, it is important to note that during the study period, there were no significant changes in the GDM diagnostic guidelines in Taiwan. Both diagnostic strategies remained accepted and available for utilization by clinicians. Nevertheless, it was observed that there was a growing trend toward the adoption of the one-step method, suggesting a potential shift in preference among healthcare professionals in Taiwan regarding GDM diagnosis [2,17,30,31].

The strength of this study lies in its large nationwide cohort design, encompassing a substantial number of participants and allowing for robust statistical analyses. The study’s long-term follow-up of participants after delivery provides valuable insights into the development of health outcomes beyond the immediate postpartum period. Moreover, the adjustment for confounding factors, such as age and parity, strengthens the validity of the observed associations [2,16,17].

However, several limitations should be considered when interpreting the results. First, the study relied on administrative data, which may be subject to coding errors or the misclassification of outcomes. Second, the study did not account for potential lifestyle factors, such as diet, physical activity, and family history, which could confound the associations. Third, the study lacked detailed information on the duration and severity of GDM, which could have provided additional insights into the risk estimates. Lastly, as with any observational study, the possibility of residual confounding cannot be entirely excluded.

The authors acknowledge that the results of the study may be influenced by the passage of time. In our previous study, we observed a significant rise in the prevalence of gestational diabetes mellitus (GDM) in Taiwan between 2004 and 2015. Over the course of 12 years, the prevalence increased from 7.6% to 13.4%, representing a 1.8-fold rise (*p*  <  0.001). This upward trend was evident across all reproductive age groups, with a particularly notable increase among women aged ≥30 years. The study also identified several risk factors associated with GDM, including advanced maternal age at conception, specific seasons of conception, urbanization level, and geographic variations [2].

Despite the increasing prevalence of GDM, our findings revealed a limited impact on adverse pregnancy outcomes. This implies that the influence of the study year itself may be relatively minimal in our current investigation. However, due to the expiration of the data retention period, further analysis at this time is not feasible. It is important for future research to delve deeper into this issue and conduct additional investigations to gain a more comprehensive understanding of the topic.

The authors also recognize that in our study, we did not specifically assess the outcomes occurring within the 120 days after delivery, which is a critical period during which certain events may still be influenced by the pregnancy itself. Regrettably, due to the expiration of the data retention period, we are currently unable to conduct further analysis to address this aspect. It is important for future research to take note of this limitation and consider examining the outcomes within the immediate postpartum period. By investigating this timeframe, a more comprehensive understanding can be gained of the potential impact of pregnancy on subsequent events.

Given the available cohort, both the propensity matching and inverse probability of treatment weighting (IPTW) methods are recognized by the authors as effective approaches to address confounding factors and enhance the robustness of the analysis. Utilizing the entire cohort for IPTW analysis could provide a broader perspective on the data. Due to the expiration of the data retention period, we were unable to proceed with propensity matching or IPTW analysis of the available cohort. Nonetheless, we will explore the potential for longer-term follow-up of the cohort in future studies.

A limitation of our study is that we did not explicitly calculate the required sample size, as typically performed in clinical trials. However, we had a large sample size, indicating a likely high statistical power. Nevertheless, it is important to acknowledge that further analysis is not possible due to the expiration of the data retention period. Thus, relying solely on the assumption of a power greater than 0.8 without conducting an actual power calculation may not be appropriate. We agree that explicitly calculating and reporting the sample size and statistical power would have strengthened our study. In future research, we are committed to conducting these calculations to ensure a more comprehensive and rigorous analysis.

The generalizability of our study results may be limited due to several factors. First, the findings should be cautiously applied to other populations or settings beyond the scope of our study due to potential variations in healthcare systems and sociocultural factors. Second, the representativeness of our cohort and the specific inclusion/exclusion criteria may restrict the applicability of the results to broader populations. Additionally, the retrospective nature of our study design and potential data limitations could impact the generalizability. Lastly, the temporal aspect of our study should be considered when interpreting the findings. Despite these limitations, future studies in diverse populations and prospective designs can further enhance the external validity of our results.

## 5. Conclusions

In conclusion, this study demonstrates that women with GDM are at an increased risk of developing T2DM, aHTN, HL, and cancer. These findings align with previous studies and highlight the importance of long-term monitoring and preventive measures in women with GDM. Further research is needed to elucidate the underlying mechanisms linking GDM to these health outcomes and explore potential interventions that can mitigate these risks.

## Figures and Tables

**Figure 1 biomedicines-11-02120-f001:**
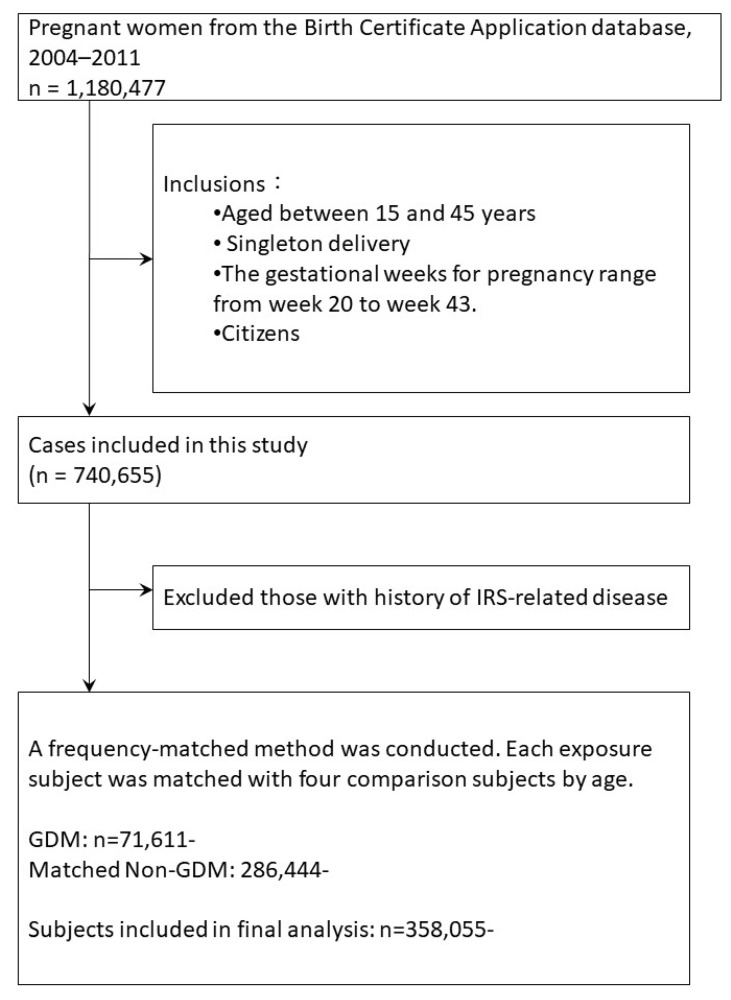
Flow chart of the data analysis process. IRS: insulin resistance syndrome; GDM: gestational diabetes mellitus.

**Figure 2 biomedicines-11-02120-f002:**
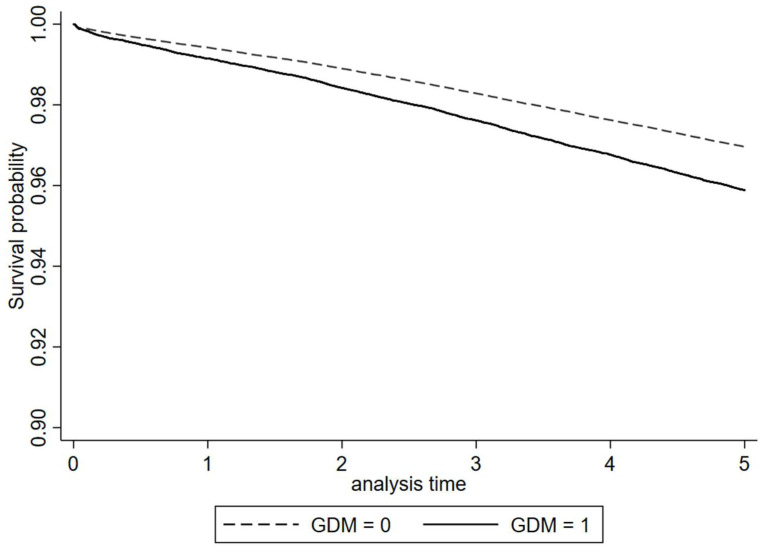
Kaplan–Meier curve for combined outcomes.

**Table 1 biomedicines-11-02120-t001:** Definitions of diseases and abbreviations with corresponding ICD codes.

	Diseases	ICD 9	ICD 10
DM	Diabetes Mellitus	250	E10–E14
aHTN	Arterial Hypertension	401–405	I10–I15
HL	Hyperlipidemia (TG or cholesterol)	272.0, 272.2, 272.4, 272.1, 272.3	E78.0, E78.2, E78.5, E78.1, E78.3
CVD	Cardiovascular disease	430–438	I60–I69
AMI	Acute myocardial infarction	410	I21
NAFLD	Non-alcoholic fatty liver disease	571.8	K76.0
PAOD	Peripheral artery occlusive disease	443	I73
CKD	Chronic kidney disease	585	N18

ICD: the International Classification of Diseases Code.

**Table 2 biomedicines-11-02120-t002:** Demographic data, *n* (%).

	GDM		*p*-Value
	Yes (*n* = 71,611)	No (*n* = 286,444)	
Age, years			
Mean ± SD	31.06 ± 4.48	31.06 ± 4.48	0.9976
15–20	894 (1.25)	3576 (1.25)	
21–25	6633 (9.26)	26,532 (9.26)	
26–30	24,360 (34.02)	97,440 (34.02)	
31–35	28,554 (39.87)	114,216 (39.87)	
36–40	9941 (13.88)	39,764 (13.88)	
41–45	1229 (1.72)	4916 (1.72)	
Comorbidity			
Cancer	478 (2.26)	5954 (1.77)	0.0064
Parity			
1	54,866 (76.62)	230,168 (80.35)	<0.0001
2	15,668 (21.88)	54,598 (19.06)	
>2	1077 (1.50)	1678 (0.59)	

**Table 3 biomedicines-11-02120-t003:** The hazard ratio of developing insulin resistance syndrome (IRS)-related diseases in relation to the baseline characteristics.

Variable	N	Event	Person-Years	Incidence *	Hazard Ratio (95% CI)	
					Crude	Adjusted **
GDM						
No	286,444	8701	1,411,250	6.17	1 (reference)	1 (reference)
Yes	71,611	2945	350,771	8.40	1.36 (1.31, 1.42)	1.36 (1.31, 1.42)
Age						
15–20	4470	58	22,238	2.61	1 (reference)	1 (reference)
21–25	33,165	635	164,358	3.86	1.48 (1.13, 1.94)	1.49 (1.14, 1.94)
26–30	121,800	3025	601,758	5.03	1.93 (1.49, 2.50)	1.93 (1.49, 2.51)
31–35	142,770	4748	702,338	6.76	2.59 (2.00, 3.36)	2.60 (2.01, 3.37)
36–40	49,705	2659	241,957	10.99	4.22 (3.25, 5.47)	4.22 (3.26, 5.48)
41–45	6145	521	29,373	17.74	6.81 (5.19, 8.93)	6.79 (5.18, 8.91)
Comorbidity	(Cancer)					
No	351,623	11,361	1,730,602	6.56	1 (reference)	1 (reference)
Yes	6432	285	31,419	9.07	1.38 (1.23, 1.56)	1.24 (1.10, 1.39)
Parity						
1	285,034	9296	1,402,640	6.63	1 (reference)	1 (reference)
2	70,266	2241	345,882	6.48	0.98 (0.93, 1.02)	0.94 (0.90, 0.98)
>2	2755	109	13,499	8.07	1.22 (1.01, 1.47)	1.07 (0.89, 1.30)

* per 1000 person-years; ** Models adjusted for age, cancer and parity.

**Table 4 biomedicines-11-02120-t004:** The hazard ratios (with corresponding 95% confidence intervals) for insulin resistance syndrome (IRS)-related diseases comparing women with GDM to those without GDM.

	GDM		Non-GDM		Hazard Ratio (95% CI)	
	Event	Incidence *	Event	Incidence *	Crude	Adjusted
All	2945	8.40	8701	6.17	1.36 (1.31, 1.42)	1.36 (1.31, 1.42)
T2DM	869	2.48	1247	0.88	2.80 (2.57, 3.06)	2.83 (2.59, 3.08)
aHTN	1004	2.86	3725	2.64	1.09 (1.01, 1.16)	1.09 (1.01, 1.16)
HL	989	2.82	3097	2.19	1.29 (1.20, 1.38)	1.29 (1.20, 1.38)
CVD	107	0.31	417	0.30	1.03 (0.84, 1.28)	1.03 (0.83, 1.28)
AMI	1	0.00	8	0.01	0.50 (0.06, 4.03)	0.52 (0.07, 4.15)
NAFLD	79	0.23	297	0.21	1.07 (0.84, 1.37)	1.07 (0.83, 1.37)
CKD	18	0.05	80	0.06	0.91 (0.54, 1.51)	0.90 (0.54, 1.50)
PAOD	38	0.11	211	0.15	0.73 (0.51, 1.02)	0.73 (0.51, 1.03)

Abbreviations used in the table: T2DM—type 2 diabetes mellitus; Ahtn—arterial hypertension; HL—hyperlipidemia (TG or cholesterol); CVD—cardiovascular disease; AMI—acute myocardial infarction; NAFLD—non-alcoholic fatty liver disease; PAOD—peripheral artery occlusive disease; CKD—chronic kidney disease; * per 1000 person-years.

**Table 5 biomedicines-11-02120-t005:** Incidence and hazard ratio of insulin resistance syndrome (IRS)-related diseases interacting with gestational diabetes mellitus (GDM) and cancer.

GDM	Cancer	N	Event	Person-Years	Incidence *	Hazard Ratio (95% CI)	
						Crude	Adjusted **
N	N	281,385	8492	1,386,490	6.12	1 (reference)	1 (reference)
N	Y	5059	209	24,760	8.44	1.38 (1.20, 1.58)	1.25 (1.09, 1.43)
Y	N	70,238	2869	344,112	8.34	1.36 (1.31, 1.42)	1.37 (1.31, 1.42)
Y	Y	1373	76	6659	11.41	1.86 (1.49, 2.34)	1.66 (1.32, 2.08)

* per 1000 person-years; ** Models adjusted for age and parity.

## Data Availability

Data is unavailable due to privacy or ethical restrictions by the Health and Welfare Data Science Center at the Ministry of Health and Welfare, Taiwan.

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
