# Peer review of "Risks after Gestational Diabetes Mellitus in Taiwanese Women: A Nationwide Retrospective Cohort Study"

_biomedicines, 2023, doi:10.3390/biomedicines11082120_

Round 1

Reviewer 1 Report

The authors study the association between GDM and future development of adverse conditions such as T2DM, hypertension, or hyperlipidemia at 5-year after delivery. The following issues can be addressed by the authors to improve the study:

1. The following are not consistent. 

Lines 98-99: "Propensity score match method was conducted. Each 1 exposure object was matched with 3 comparison object by age."

Line 124-125 "after matching GDM women with non-GDM women by age with an 1:4 manner, there were 71,611 GDM women and 585,695 non-GDM women included."

Table 1

Please clarify whether the matching was 1:4 or 1:3. Please clarify why the count of non-GDM women was 585,695 and later 286,444

2. Table 2 presents the hazard ratio of developing related disease, but "related disease" are not defined. This is not mentioned in the abstract.

3. Table 4 Correct the title. The table does not present the association between GDM and cancer.

4. Specify the time period for identifying the adverse conditions. Is it between 2004 and 2011?

5. Be consistent when referring to adverse outcomes: Table 4 "various health outcomes," Table 2 "related disease."

6. Add a paragraph to explain the GDM diagnosis criteria in Taiwan during the study period. Were there any changes in the guidelines?

7. Have you studied whether the results are impacted by time. Can you provide GDM rates by year during the study period? When adding year to the models does it have a significant effect.

8. Can you add Kaplan-Meier curves for each and combined outcome to supplementary materials?

9. Have the authors explored defining the outcomes starting with 120 days after delivery? Events occurring in the 120 days after delivery could be pregnancy related. 

10. Lines 91-97: Cohort and group creation is unclear. Please see below several issues.

Were there women with both GDM and non-GDM pregnancies? If yes, were they included in both groups? Please clarify how these cases were treated.

Why for GDM cases the index date is the last delivery date and for non-GDM cases the index date is the first delivery date?

I suggest to adopt the following approach. Consider all women who delivered between 2004-2011 (after applying the necessary exclusion: age, singleton delivery, etc) and define the index date as the first delivery date during the study period. Based on the index delivery classify women into GDM or non-GDM cases. You can proceed with propensity matching or inverse-probability of treatment weighting (IPTW). 

11. Lines 120-128: The drop from 1,180,477 to 71,611 GDM women and 585,695 non-GDM women is high. 50% of the sample is lost. Please explain and include a flow chart indicating how many women are lost for each exclusion.

12. You could attempt IPTW to use the entire cohort you have.

Reviewer 2 Report

Manuscript ID: biomedicines-2477911

Title: Risk of developing type 2 diabetes, hypertension and dyslipidemia 5-year follow-up after delivery in women with GDM: a nationwide cohort study in Taiwan

The primary arm of this study is to investigate the risk of developing type 2 diabetes mellitus (T2DM), hypertension (HTN) and hyperlipidemia (HL), 5- year follow-up after delivery in women with gestational diabetes mellitus (GDM) using a nationwide cohort study in Taiwan. The second objective of this study is to investigate the risk of developing cardiovascular disease (CVD), Non-alcoholic fatty liver diseases (NAFLD) and chronic kidney disease (CKD) 5-year follow-up after delivery in women with GDM.

Comments and Suggestions for Authors: The manuscript is an interesting study, but requires some considerations.

Title: The title only refers to the primary objectives of the study. It could go global. The type of study design (retrospective cohort) should be indicated. It would be better not to use acronyms (GDM) in the title.

Abstract: The acronym IQR, although widely used, should be defined. Line 24. Hypertension should be specified as Arterial hypertension. This should apply to the entire manuscript.

1. Introduction: Bibliographical references must be presented numerically between square brackets. This is applicable to the rest of the manuscript.

2. Material and methods: It should be indicated how the diagnosis of GDM or T2DM was reached. Acronyms that have already been introduced previously are redefined and terms without acronyms that have already been defined are repeated again. This must be corrected throughout the manuscript.

Line 88. “Diabetes mellitus” should be changed to pregestational diabetes mellitus.

Line 94. “The index date for the exposure group was the latest delivery day after her GDM diagnosis no matter how many parities she had before or after the GDM diagnosis. The index date for the comparison group was the 1st delivery day”. Could it be a bias?

Line 98. “Each 1 exposure object was matched with 3 comparison object by age”. However, Line 125 states that "matching GDM women with non-GDM women by age with an 1:4 manner". Please clarify it. Indicate the number of participants with missing data for each variable considered.

3. Results:

Line 125. It states that "there were 71,611 GDM women and 585,695 non-GDM women included". However, in the Tables these proportions were YES (n=71,611) No (n=286,444). Please clarify this throughout the manuscript, including the Abstract section. All Tables should include a footer explaining the abbreviations used.

Table 1. It is not clear if comorbidity cancer presented was prior to the follow-up of the cohort. If so, shouldn't these patients have been excluded from the study?

Table 3. DM should be changed to T2DM.

Table 3. Results are given for variables such as AMI and PAOD, which are not proposed in the study objectives, nor are they presented in the Material and methods section.

Table 4. The title of the table should summarize your presentation. The second paragraph should be left for the text.

Table 4. No data shown for NAFLD.

4. Discussion: The generalisability (external validity) of the study results should be discussed.

Line 235. Diabetes mellitus should be changed to T2DM. Consider this throughout the manuscript.

The authors honestly acknowledge some limitations of the study. Most impossible to fix. The calculation of the necessary sample size is not indicated

Minor editing of English language required

Round 2

Reviewer 1 Report

Thank you for answering my previous comments. 

The paragraph on lines 286-301 was added to describe the diagnostic criteria for GDM in Taiwan during the study period. It would make the description clearer if you could add the thresholds for GDM diagnostic in this paragraph. Also, if available please add the percentage of women diagnosed based on the one-step method and on the two-step approach.

Reviewer 2 Report

Authors make changes to the manuscript that improve it.

Author Response

The authors truly appreciate your valuable comments. Thank you so much.